# Amplitude-Integrated EEG Monitoring in Pediatric Intensive Care: Prognostic Value in Meningitis before One Year of Age

**DOI:** 10.3390/children9050668

**Published:** 2022-05-05

**Authors:** Jonathan Beck, Cecile Grosjean, Nathalie Bednarek, Gauthier Loron

**Affiliations:** 1Department of Neonatology, Reims University Hospital Alix de Champagne, 51100 Reims, France; jbeck@chu-reims.fr (J.B.); cecile-grosjean@live.fr (C.G.); nbednarek@chu-reims.fr (N.B.); 2CReSTIC EA 3804 UFR Sciences Exactes et Naturelles, Campus Moulin de la Housse, Université de Reims Champagne Ardenne, 51100 Reims, France

**Keywords:** meningitis, cerebral monitoring, seizure, amplitude-integrated electroencephalography, neurological outcome, pediatric critical care, infant

## Abstract

Pediatric morbidity from meningitis remains considerable. Preventing complications is a major challenge to improve neurological outcome. Seizures may reveal the meningitis itself or some complications of this disease. Amplitude-integrated electroencephalography (aEEG) is gaining interest for the management of patients with acute neurological distress, beyond the neonatal age. This study aimed at evaluating the predictive value of aEEG monitoring during the acute phase in meningitis among a population of infants hospitalized in the pediatric intensive care unit (PICU), and at assessing the practicability of the technique. AEEG records of 25 infants younger than one year of age hospitalized for meningitis were retrospectively analyzed and correlated to clinical data and outcome. Recording was initiated, on average, within the first six hours for *n* = 18 (72%) patients, and overall quality was considered as good. Occurrence of seizure, of status epilepticus, and the background pattern were significantly associated with unfavorable neurological outcomes. AEEG may help in the management and prognostic assessment of pediatric meningitis. It is an easily achievable, reliable technique, and allows detection of subclinical seizures with minimal training. However, it is important to consider the limitations of aEEG, and combinate it with conventional EEG for the best accuracy.

## 1. Introduction

Acute meningitis is a major health problem around the world. Regardless of age, its global prevalence exceeded 10 million cases each year [1]. In 2017, considering infants under one year of age, incidence was estimated around 0.11% [2]. In newborns, incidence ranges from 0.22 to 0.5 per 1000 live births, even higher in premature and low birth weight infants, being at 20.4% and 13.6%, respectively [3,4,5,6].

In France, from 2010 to 2014, the predominant bacterial species were Streptococcus agalactiae (57.2%) and Escherichia coli (16.8%) in infants less than two months of age. Streptococcus pneumoniae (44%) and Neisseria meningitidis (31.7%) were predominant in infants between 2 and 12 months of age.

The mortality rate for bacterial meningitis in children is around 6.6% overall, and 7.2% under one year [7]. Neonatal meningitis mortality accounted for 6% to 8% of overall causes of death in children under five years of age in 2015 [8].

Because of the high risk of morbidity and death caused by meningitis, its diagnosis and treatment are an emergency. The risk factors for mortality are multiple: the severity of the infection, delay in diagnosis, prematurity, low weight for gestational age, seizures, and mechanical ventilation [4,5,9,10]. Furthermore, due to the disease and its complications, up to one third of survivors will develop permanent sequelae, such as epilepsy; encephalopathy; motor or cognitive disorders; neurosensory sequelae (deafness, visual disorders); learning disabilities; and behavioral, language, or coordination disorders [4,11,12]. Brain imaging is performed to detect complications such as cerebral abscesses, ventriculitis, hydrocephaly, or strokes [4,10]. Imaging abnormalities are one of the risk factors for developing neurological sequelae [12].

The symptoms suggestive of meningitis vary according to age and the germ involved. Typically, the younger the child is, the less specific the signs will be. Digestive symptoms, irritability, or lethargy should suggest the diagnosis, but fever or a bulging fontanel may be absent in half or two thirds of cases, respectively [13,14]. Any delay in the start of suitable antibiotics will result in poorer outcome [15,16]. In most cases, meningitis is revealed by markers of severity or even complications such as seizures in approximately 20–30% of patients [6,9,13,17]. Reported risk factors for seizures are meningitis due to Streptococcus pneumoniae, imaging abnormalities, and immunosuppression [18]. Seizures are associated with in-hospital mortality and poor outcome [18]. However, seizures may be predominantly subtle or subclinical, especially among the youngest, which is why paraclinical examinations, such as brain monitoring, are helpful for early recognition.

The electroencephalogram (EEG) is the gold standard for detecting and characterizing seizures or epilepsy, and should be used in all cases of observed or suspected seizures [19,20]. However, continuous EEG monitoring is not available at any time in every neonatal or pediatric intensive care unit. Indeed, it requires trained and specially-skilled caregivers to accurately set the complex equipment, and then to interpret the full wealth of the trace. In order to cover all of the needs, the examinations are often limited from 30 to 60 min [20,21]. To enhance these accurate, but frequently intermittent, records, clinicians need trends that they can analyze continually.

The amplitude EEG (aEEG) was developed in the late 1960s by Douglas Maynard at the London Hospital, United Kingdom, for continuous monitoring of brain activity in adult, resuscitated patients [22]. Through amplitude-integrated EEG processing, the raw-EEG signal from one or two derivations is filtered, rectified, sampled, and then displayed at a slow speed (around 3.5 h per screen width). AEEG emphasizes the amplitude change of low frequency wavelengths over time. With minimal training, aEEG can be set and interpreted by a caregiver at bedside [23]. The recognition of background pattern and electrical seizures is quite good, as well as inter-observer reliability [21,23,24]. An extensive literature has pointed out the many advantages of aEEG in monitoring neonatal encephalopathy, neonatal seizures, and preterm birth. It has shown some value in older children, admitted for convulsive disorder, circulatory arrest, or cardiac surgery [19,25,26,27,28,29,30,31,32,33,34,35].

The relevance of aEEG in infectious meningitis in infants has not been reported yet.

This study aimed at assessing the predictive value of aEEG monitoring in infectious meningitis among children under one year of age. We hypothesized that it may help for early prediction of the neurologic outcome.

## 2. Materials and Methods

### 2.1. Study and Population

The aEEG was introduced in our pediatric and neonatal intensive care units in 2004, complementary to the use of standard EEG. We primary use aEEG when the EEG is not available (i.e., at nights and weekends), and for brain monitoring over days.

The present study was monocentric, observational, and took place over a period of 10 years. We retrospectively included patients with the following criteria: infants less than one year old; hospitalized between 1 January 2004 and 31 December 2014, in neonatal or pediatric intensive care units at the tertiary University Hospital of Reims with infectious meningitis; and available aEEG recording. During this period, the neonatal and pediatric intensive care units were combined into one polyvalent unit, which made the collection of records easier, and after 2017, due to the introduction of mandatory vaccination of children in France, the number of cases included could have been smaller.

### 2.2. Demographic, Clinical, and Biological Data

The Medical Information Department screened patients meeting the inclusion criteria. Demographical data extracted from medical charts included: sex, age (in days), weight (kg), height (cm), head circumference (cm), and history of neurological pathology (epilepsy, febrile seizure, encephalopathy, other).

Data characterizing meningitis were: neurological signs (tone, consciousness, bulging fontanel, seizures), fever, altered general condition, vomiting or feeding impairment, respiratory or hemodynamic distress, time from onset of symptoms to diagnosis of meningitis, inflammatory markers (CRP (mg/L), PCT (ng/mL)), initial leukocytes (/mm^3^)), bacteriological tests (blood culture, cerebrospinal fluid test with cell count, protein, glucose, lactate, and culture), and antimicrobial therapy (type, start-up time, and duration).

Data concerning outcome were death, diagnosis of epilepsy, subsequent encephalopathy (yes/no), radiological brain injuries, and duration of hospital and PICU stay. The Pediatric Cerebral Performance Category (PCPC) was assessed one year after hospital discharge, according to Utstein’s guidelines [36,37,38,39]. This scale is used to categorize the child’s deficit and disability in six levels: 1/normal, 2/mild disability, 3/moderate disability, 4/severe disability, 5/vegetative state or coma, 6/death. For statistical purposes, PCPC scores were categorized as “favorable outcome group” (scores 1–2–3) versus “unfavorable outcome group” (PCPC score 4–5–6). The PCPC score presented in this study is available in Appendix A.

### 2.3. aEEG Data

The aEEG traces were recorded with one of these devices: Olympic CFM^®^6000 (three electrodes), Olympic Brainz Monitor^®^, or Brainz BMR3 Monitor^®^, and read on the device or analyzed using the “Analyze Research” software for Brainz Monitors^®^ (all those devices and software are distributed by Natus^®^, Middleton, WI, USA).

aEEG traces were characterized as follow: duration (min), quality of the recordings, lower and upper margin, presence of modulation, presence and number of electrical seizures, time from beginning to first continuous trace, and time from beginning to first modulation. Background pattern was classified according to the classification popularized by Hellström-Westas as: continuous normal, discontinuous, burst-suppression, continuous low-voltage, flat trace [40].

The background was scored from 1 to 4 points, for every 3-h period, for the first 24 h of recording, as previously reported [41]. Normal trace, discontinuous trace, burst-suppression, and low-voltage trace and flat trace accounted for 1, 2, 3, and 4 points, respectively. The score could therefore range from 8 to 32, with higher scores reflecting a more impaired background activity. If the 24 h period was not complete, the score of the last recorded period was repeated for the following missing periods.

Electrical seizures on aEEG were acknowledged by notches on the amplitude trace and evocative, rhythmic pattern on the continuous EEG.

Appendix A illustrates an example of seizure on the aEEG trace used in this study, and Appendix A is an example of artifacts caused by ventilation. They are available in Appendix A.

Recording quality was evaluated according to the impedance value and its variations.

Each trace was interpreted by a pediatric resident recently trained in aEEG interpretation (JB), and blindly reviewed by a hospital clinician experienced in aEEG interpretation (GL).

### 2.4. Statistics

Statistical analyses were performed using STATA 16 software (Stata/SE version, STATACorp LLC, 4905 Lakeway Drive, College Station, TX 77845-4512, USA). Descriptive statistics were expressed using standard deviation ± means for quantitative variables, and percentage (%) for qualitative variables. Univariate analyses were performed with Student’s T test for quantitative variables, and chi-squared test or Fisher’s exact test for qualitative variables. The strength of statistical association was performed by measuring the odds ratio and its 95% confidence interval (CI) for the predictivity of neurological prognosis by aEEG items. The predictive value of the background trace obtained by aEEG during the first 24 h of recording was plotted on an ROC curve with the 95% CI of it. Threshold for significance was defined as *p* < 0.05.

## 2.5. Ethics Conflict of Interest

There are no conflicts of interest to report. The present study was declared to an independent French administrative authority named “Commission Nationale de l’Informatique et des Libertés” (CNIL), in 5 October 2016 under the declaration number 1995487 v0.

## 3. Results

### 3.1. Population and Epidemiological Characteristics

We identified 35 patients less than 1 year old, admitted to the pediatric and neonatal intensive care units with acute meningitis, whose medical charts reported an aEEG monitoring. For 10 children, the original aEEG trace was unusable because of technical issues or missing data. The demographic characteristics of these 10 patients did not differ from the subsequently included patients (data not shown) (Figure 1). Of the 25 children finally included in the study, 18 of them (72%) were classified in the favorable outcome group (PCPC 1–2–3), and 7 (28%) in the unfavorable outcome group (PCPC 4–5–6) (Table 1).

Two patients presented with a previous neurological history: one had hydrocephalus, and the second had spina bifida.

### 3.2. Characteristics of Meningitis and Hospitalization

The mean time from symptom onset to diagnosis was 38.65 ± 33.48 h for 80% of the patients (*n* = 20). For the remaining 20% (*n* = 5), the time was not clearly indicated, but was less than 12 h for each of them. The mean time between the onset of symptoms and the start of anti-microbial therapy was 23.2 ± 15.75 h for 80% of patients (*n* = 20). For 16% (*n* = 4) of the patients, the time was not clearly indicated, but was less than 12 h for each patient, and the time was not specified for 4% of the patients (*n* = 1).

Patients in the unfavorable outcome group had a significantly higher initial PCT than the favorable outcome group. The unfavorable outcome group had a significantly higher cerebrospinal protein concentration on the first lumbar puncture.

Mean time for parenteral nutrition weaning and the length of stay in PICU were significantly increased in the unfavorable outcome group. The meningitis and hospitalization characteristics of the cohort are described in Table 2.

### 3.3. Patient Outcome

Two (8%) patients died. Twelve patients (52%) developed clinical neurological and sensorial sequelae. Three patients (13%) were diagnosed with epilepsy, and one (4%) with encephalopathy at discharge. Brain injuries were reported for 16 patients (64%), divided into ventricular enlargement (*n* = 5 (20%)), multifocal injury of the parenchyma (*n* = 3 (12%)), extracerebellar collection (*n* = 3 (12%)), and brain injuries involving several of these locations (*n* = 5 (20%)).

### 3.4. Description and Prognostic Value of aEEG

Median time to start aEEG recording was 3 ± 5.5 h after the beginning of the PICU hospitalization, for a mean duration of 99.34 ± 83.21 h, with a significant difference between the two groups, with longer recordings for the unfavorable outcome group with 161.43 ± 102.29 versus 75.19 ± 62.27 h (*p* = 0.016). Distribution of the aEEG recordings over time is presented in Figure 2. However, there was no significant difference between the two groups regarding the quality of the record (unfavorable group 83.34 ± 22.60 versus favorable group 84.01 ± 21.91%, *p* = 0.946). Furthermore, we noted that aEEG recordings were of sufficient quality to be interpreted for 84% of the recordings.

Looking for seizures, 202 notches were reported and analyzed on aEEG traces for the 25 patients. Forty-eight (24%) notches were considered as artifacts. Diagnosis of electrical seizure was validated on aEEG for 154 (76%) notches, accounting for 11 patients overall (44%). Among them, two infants (18%) presented electrical seizures only (four seizures each). Nine patients (82%) presented both clinical and infraclinical seizures during recordings. They presented a median number of 5 ± 12 electrical seizures only (from 2 to 69 seizures). Overall, 18 infants (72%) presented clinical or electrical seizures at any moment of their disease.

Seizures and electrical status epilepticus were more frequent in the unfavorable outcome group compared to the favorable outcome group. The presence of electrical seizures and electrical status epilepticus was significantly observed within the unfavorable outcome group.

The first 24-h background pattern score was significantly higher in the unfavorable outcome group compared to the favorable outcome group.

We did not document a significant difference for the “recovery of a normal background” or the “recovery of a modulation”, whose details are presented in Table 3. The predictive value of the background score at aEEG is presented by the ROC curve in Figure 3, with an area under the curve of 0.758 (95% CI 0.521–0.995). The cut-off point for the best sensitivity–specificity “couple” is a background tracing score of 11; when it is ≥ 11, the sensitivity was 71.43% and the specificity was 66.67% for an unfavorable prognosis by the PCPC scale (4–6).

## 4. Discussion

To the best of our knowledge, this is the first report on the contribution of aEEG in meningitis in infants under one year of age. Brain monitoring by aEEG in meningitis is easy and helpful to prognosis.

We noticed the ability of aEEG monitoring in identifying subclinical seizures in this setting. Moreover, the presence of electrical seizures, status epilepticus, and a higher 24-h background pattern score were associated with an unfavorable outcome.

The impact of seizures during acute meningitis on neurologic outcome has already been highlighted. If seizures are prolonged, difficult to treat, or develop 72 h after admission, neurologic sequelae are more likely to occur, and could be suggestive of a cerebrovascular event [16]. Ouchenir et al. showed that the presence of seizures is a prognostic marker of motor sequelae, developmental delay, and death [10]. Early detection and treatment of seizures is associated with a higher likelihood of seizure cessation [42]. AEEG could allow earlier detection of seizures, particularly subclinical ones, for a more reactive initiation of therapy, and to prevent them from persisting. Moreover, a recent study showed that the time elapsed from ICU admission to initiation of continuous EEG was associated with increased mortality [43]. A dedicated study would be needed for aEEG in patients with meningitis.

Here, electroencephalographic monitoring using aEEG documented a high prevalence of seizures in meningitis among infants, especially with few to no clinical signs in meningitis. Eighty-eight percent of seizures recognized at aEEG had no clinical substrate in the medical charts in our study. The retrospective design of this work and the absence of contemporary video recording did not help in identifying all clinical manifestations. However, the high prevalence of subclinical seizures in pediatrics in acute care has already been highlighted [44]. AEEG is even more accurate in the detection of infra-clinical status epilepticus than infra-clinical isolated seizures [26].

Compression of the trace, due to the amplitude-integrated processing, may limit the ability to detect seizures. In our experience, assessing the raw EEG when a notch is visualized on the aEEG background pattern is a crucial step which increases the probability of detecting single short seizures [45]. AEEG dedicated devices can display aEEG and raw EEG simultaneously. Bruns et al. reported a better seizure detection on aEEG rather than clinical observation, especially for non-convulsive status epilepticus [31]. Shellhaas et al. showed that aEEG did not increase the number of neuroimaging examinations nor the use of antiepileptic drugs, but that it allowed less overtreatment of children based only on clinical suspicions of seizures without electroencephalographic confirmation [29].

The clinical course and prognosis of meningitis observed in our study is in agreement with other studies on the subject. The mortality rate found in our study is comparable to that described in the literature among infants [7]. However, we reported fewer sequelae in the present work (16%) than the literature reported [4,11,12]. This can be explained by the limited population in our study. Yet, long term epilepsy occurred in 13% of our population, close to the observation of Ben Hamouda et al. and also Briand et al., during the follow-up at around 24 months [4,11].

Beyond meningitis, aEEG monitoring has proven prognostic accuracy in pediatrics and among adults. Thomas et al. showed that a moderately or severely altered background pattern, according to the classification of al Naqeeb [46], in the first 24 h of aEEG monitoring was associated with a prognosis of death for a positive predictive value of 77.8% [47]. In a more recent study (Bourgouin et al.), the background pattern during the first 24 hours after cardiac arrest was significantly associated with outcome [41]. This is corroborated in adults after cardiorespiratory arrest, with a poor prognosis in 66% of patients with a flat tracing in normothermia [48,49].

Poorly reactive background pattern on conventional EEG monitoring is associated with an adverse outcome [50,51]. In our study, we found a significantly higher 24-h background score in patients with an unfavorable neurological prognosis according to the PCPC, in agreement with other studies. A significant improvement of the background pattern was observed in the study of Gui et al. in infants who had been through or would go through a congenital heart disease surgery, as well as sleep–wake rhythm abnormalities associated with severe infections [52]. After cardiac surgery in infants less than three months of age, a lack of postoperative sleep–wake cycling predicted poorer motor and cognitive outcomes at one year of age [53]. Due to the heterogeneity in duration and timing of aEEG recordings, our study was probably not suitable to assess sleep–wake cycling prognosis value. This could be evaluated in the future.

One of the potential interests of aEEG is its ease of use. In the present work, we observed that aEEG was initiated within the first 6 h for 72% of patients. In pediatric and neonatal intensive care units, time to electroencephalography has been correlated to mortality [43]. This should be assessed using aEEG in a dedicated study. The aEEG monitor is easy to set up and quickly available, even out of care hours, unlike the standard EEG in many centers [19]. In our day practice, we found that aEEG is a reliable and reproducible technique, and the aEEGs recordings were of sufficient quality for interpretation more than 80% of the time. Moreover, it can be used over long periods of time, as 56% of the patients were recorded for more than 72 h with a good quality of recording, corroborated by previous works [30,54]. However, aEEG has several limitations. In current dedicated devices, aEEG is recorded over one or two channels, whereas conventional EEG uses 10 to 20 channels by default. This results in poor spatial resolution, and less sensitivity to very localized phenomenon. Underestimation therefore seems inevitable, especially with single-lead monitors with an electrical seizure detection rate of about 40%. A two-lead monitor, as used in 80% of cases in our study, improves the sensitivity of detection of electrical seizures [55]. Kobayashi et al. associated the aEEG with the “spectral edge frequency” technique to increase the detection of seizures by staining the tracing [25].

AEEG monitoring requires training for setting and interpretation. A defective contact of the electrodes or a bad impedance makes the trace uninterpretable. In addition, misinterpretation of traces due to artifacts or lack of training can lead to an under- or overestimation of electrical seizures, and even an overuse of antiepileptic treatment [19,56]. According to Bruns’ study, non-expert recognition of seizures showed good sensitivity, but the high false-positive rate highlights the need for experts used to raw and conventional EEG [33]. Patient movements and care must be annotated on the trace because they induce artifacts that may lead to misinterpretation. In addition to seizures, left/right asymmetry of background can be interpreted relatively safely after a short training and indication of artifacts on the tracing [57,58]. However, the questions of how to interpret the background pattern in patients beyond the neonatal period and how to handle this interpretation arise. Indeed, there are various classifications applied to premature and newborns [46,59,60,61], but to date, there is no similar one for older patients [33].

The population of this study included patients from the neonatal period to one year of age, even if the diagnostic and therapeutic management was a little different between these two populations. We have therefore pooled them in this work due to the common susceptibility of their developing brain to produce seizures, and because they were hospitalized in the same intensive care unit.

Finally, our study had some limitations that may have somewhat impacted the validity of our results. It was a retrospective study with a small number of patients, and all of them required hospitalization in an intensive care unit, which may not be representative of the entire population of pediatric patients with meningitis. We can also assume that not all clinical seizures were recorded in the medical records, providing an information bias, which may have led to an overestimation of the number of subclinical seizures.

## 5. Conclusions

In conclusion, according to our results, and in agreement with the studies previously carried out on the use of the aEEG and the studies concerning meningitis, the aEEG is a promising and useful neurological monitoring tool during the evolution of severe meningitis in infants. The technique is reproducible, easy to set up, and reliable with minimal training in the interpretation of the tracings and the detection of electrical seizures. It should be used as soon as the diagnosis is established and from the beginning of care. This could improve management and allow a better assessment of the prognosis of severe meningitis in view of the statistically significant association found between the presence of seizures or convulsive state and a poor neurological prognosis.

However, the question remains whether a more reactive management of seizures would improve the prognosis of these patients. Brain monitoring with aEEG, offering a more comprehensive inventory of seizures, could be considered in a prospective work to provide some elements of the answer.

## Figures and Tables

**Figure 1 children-09-00668-f001:**
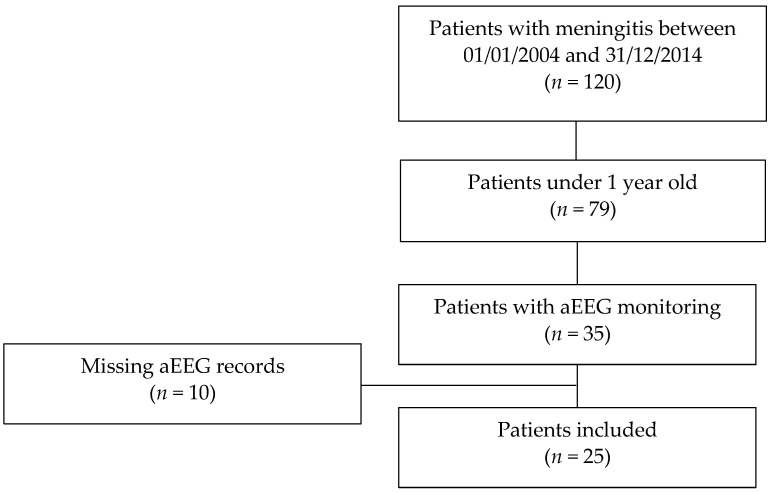
Flow chart of the study. Thirty-five patients met the inclusion criteria: hospitalization in pediatric or neonatal intensive care unit; age: less than one year; aEEG monitoring reported in medical charts. Ten patients were subsequently excluded due to missing the complete aEEG dataset.

**Figure 2 children-09-00668-f002:**
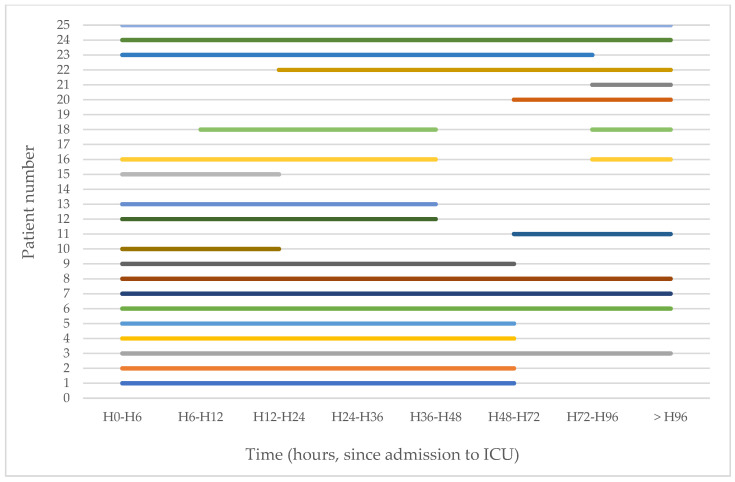
Distribution over time of aEEG recordings for each patient (1–25). Patients 14, 17, and 19: aEEG was recorded beyond H96 of hospitalization in ICU.

**Figure 3 children-09-00668-f003:**
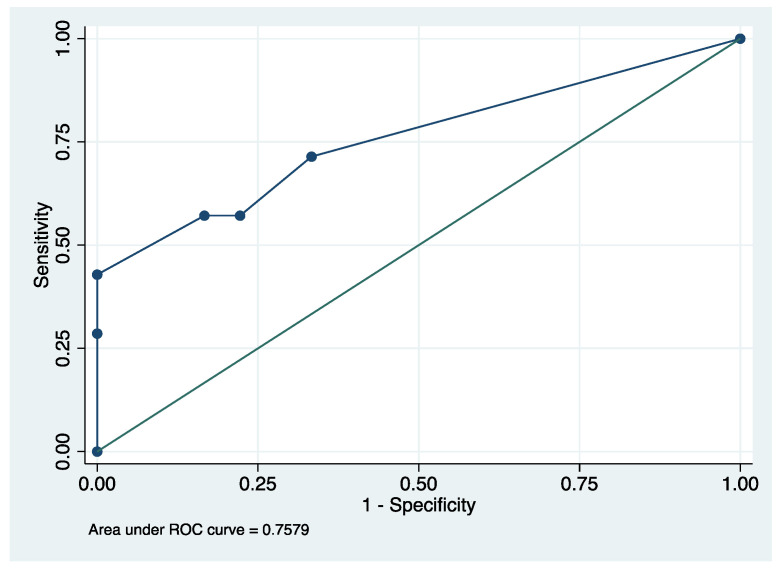
Predictive value of the background score at aEEG, ROC curve.

**Table 1 children-09-00668-t001:** Characteristics of patients included in the study, according to outcome scores.

Patient Characteristics	Favorable Outcome Group *n* = 18	Unfavorable Outcome Group *n* = 7	*p*
Sex ^1^, men	10 (55.6)	2 (28.6)	0.378
Age ^3^, days	22 ± 60	61 ± 86	0.471
Weight ^2^, g	3732 ± 1494	4187 ± 2837	0.602
Size ^2^, cm	51.6 ± 7.0	52.9 ± 9.4	0.725
Head circumference ^2^, cm	35.6 ± 4.5	36.2 ± 5.5	0.795
Center referring the patient to intensive care ○ University hospital ^1^ ○ Other hospital ^1^	6 (33.3) 12 (66.7)	1 (14.3) 6 (85.7)	0.626
Neurological history ^1^	2 (11.1)	0 (0)	1

^1^ Data expressed in *n* (%). ^2^ Continuous variables expressed as averages ± standard deviation. ^3^ Variables expressed in median ± interquartile range.

**Table 2 children-09-00668-t002:** Bacteriological, clinical, and management characteristics of recorded patients, according to outcome.

Characteristics ^1^	Favorable Outcome Group *n* = 18	Unfavorable Outcome Group *n* = 7	*p*
Symptoms ^1^			
Neurologicals exclusive *	1 (5.6)	1 (14.3)	
Generals exclusive **	6 (33.3)	1 (14,3)	0.518
Both	11 (61.1)	5 (71.4)	
Time from symptoms onset to diagnosis ^2^	37.1 ± 35.8	45 ± 24.7	0.683
<12 h ^1^	5 (27.8)	3 (42.9)	
12–24 h ^1^	5 (27.8)	2 (28.6)	0.853
>24 h ^1^	8 (44.4)	2 (28.6)	
Initial CRP (mg/mL) ^2^	78.1 ± 101.5	150.9 ± 118.9	0.138
Initial PCT (ng/mL) ^2^ (0 ^3^/4 ^3^)	25.8 ± 22.8	117.3 ± 48.1	<0.001
Initial leukocytes (/mm^3^) ^2^	9.0 ± 7.6	6.4 ± 5.1	0.414
Positive blood culture ^1^	12 (66.7)	5 (71.4)	0.819
First lumbar puncture ^1^	17 ***	7	-
Leukocytes (/mm^3^) ^2^ (3 ^3^/0 ^3^)	2055.9 ± 2341.1	4135.9 ± 7357.4	0.323
Hematite (/mm^3^) ^2^ (4 ^3^/1 ^3^)	2829.1 ± 4978.5	4963.7 ± 3049.4	0.459
Proteins (g/l) ^2^ (3 ^3^/0 ^3^)	2.41 ± 1.57	5.08 ± 2.60	0.007
Glucose (g/l) ^2^ (3 ^3^/0 ^3^)	1.39 ± 1.13	0.62 ± 0.96	0.138
Lactates (mmol/l) ^2^ (8 ^3^/4 ^3^)	6.54 ± 4.28	10.53 ± 2.22	0.156
Chlore (mmol/l) ^2^ (4 ^3^/3 ^3^)	115.5 ± 5.8	109.3 ± 8.4	0.104
Cerebrospinal fluid sterile ^1^ Positive bacteriology ^1^	2 15	0 7	
○ Streptococcus B ○ Pneumococcus ○ Enterovirus ○ Enterococcus faecalis ○ Escherichia Coli K1 ○ Meningococcus B ○ Coagulase negative staphylococci	4 (22.2) 5 (27.8) 1 (5.5) 1 (5.5) 2 (11.1) 1 (5.5) 1 (5.5)	2 (28.6) 2 (28.6) 0 0 3 (42.9) 0 0	
Time from initial symptoms to treatment ^1^			
<6 h	5 (27.8)	2 (28.6)	
6–12 h	4 (22.2)	2 (28.6)	1
>12 h	8 (44.4)	3 (42.9)	
Total duration of antibiotic ^2, 4^	16.7 ± 8.1	20.3 ± 6.6	0.305
Mean time of parenteral nutrition weaning ^2, 4^ (1 ^3^/3 ^3^)	4.3 ± 7.1	13.6 ± 12.5	0.043
Length of stay in PICU ^2, 4^	7.9 ± 4.7	20.4 ± 5.9	<0.001
Total length of stay at hospital ^2, 4^	22.4 ± 22.1	33.0 ± 12.7	0.247

^1^ Data in *n* (%). ^2^ Continuous variables expressed as averages ± standard deviation. ^3^ N missing data. ^4^ Data expressed in days. * The patient in the favorable outcome group had a bulging fontanel; this one in the unfavorable outcome group had seizures. ** Patients had fever, vomiting or feeding impairment, purpura, respiratory or hemodynamic distress. *** One patient did not undergo a lumbar puncture because of an exudative dysraphism, but had a positive blood culture for streptococcus B.

**Table 3 children-09-00668-t003:** Prognostic value of aEEG features during recording.

aEEG Characteristics	Favorable Outcome Group (*n* = 18)	Unfavorable Outcome Group (*n* = 7)	*p*	OR (IC 95%)
Background score over first 24 h ^2^	10 ± 3	16 ± 7	0.008	6.66 (0.95–46.56)
<161	15 (83.3)	3 (42.9)	0.066	
≥161	3 (16.7)	4 (57.1)		
Electrical epileptic seizures ^1^				15.60 (1.48–164.38)
Yes	5 (27.8)	6 (85.7)	0.021	
No	13 (72.2)	1 (14.3)		
Electrical status epilepticus ^1^				20 (2.21–180.90)
Yes	2 (11.1)	5 (71.4)	0.007	
No	16 (88.9)	2 (28.6)		
Recovery of a modulation ^1^				
Yes	15 (83.3)	5 (71.4)	0.597	0.5 (0.06–3.91)
No	3 (16.7)	2 (28.6)		
Recovery of a normal background ^1^				
Yes	11 (61.1)	1 (14.3)	0.073	0.11 (0.01–1.08)
No	7 (38.9)	6 (85.7)		

^1^ Data in *n* (%). ^2^ Continuous variables expressed as averages ± standard deviation.

## Data Availability

Data available on request due to restrictions, e.g., privacy or ethical. The data presented in this study are available on request from the corresponding author.

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
