# Peer review of "Amplitude-Integrated EEG Monitoring in Pediatric Intensive Care: Prognostic Value in Meningitis before One Year of Age"

_children, 2022, doi:10.3390/children9050668_

Round 1

Reviewer 1 Report

Thank you for the opportunity to review your manuscript. It is a novel concept, good methodology, well written with noted limitations & does not draw & long conclusions.

My only small points of clarification for this retrospective study is why were aEEGs being recorded in the first place for this cohort of patients and of the 10 patients missing aEEGs, was there a bias within this group eg too unwell to have aEEGs recorded, too well to consider the need for an aEEG or something different?

Author Response

We thank the reviewer for her/his valuable comments. 

Regarding the aeeg vs eeg usage policy, the aEEG was introduced in our pediatric and neonatal intensive care units in 2004, complementary to the use of standard EEG. We primary use aEEG when the EEG is not available (i.e., at nights and week-ends) and for brain monitoring over days. We have added this explanation in the "methods" to clarify this point. 

Concerning the 10 patients with missing aEEG data, they were not different from the study population. They did benefit from aEEG recordings, reported in the medical charts, but the original tracings were not found, probably because of the retrospective nature of this work. This was explained in the first paragraph of the results.

An updated manuscript has been submitted.

Thank you again for your valuables comments, 

Sincerely Yours.

Reviewer 2 Report

This is interesting research especially for neonatologists and intensivists. It is very interesting to introduce aEEG to PICU for detecting asymptomatic or early convulsions in bacterial meningitis in infants.  These are my suggestions for a better explanation of the tables and figures.

Figure 1 showing flow chart: each figure must have a full description about what it shows, so that it is clear to the reader as a stand-alone unit.

Table 1. The name of the table must clearly and in detail explain what the table shows. The characteristics of the patient should not be explained in the title of the table; that part should be separated and explained in the text.

Figure 2.The name of the figure must clearly and in detail explain what it shows; with the ordinate and the abscissa, it should be written the number of patients and the chronological sequencein hours, not in the title of the figure.In supplement materials: abbreviations should be avoided in table names

Author Response

We thank the reviewer for its interest in this report and for his/her remarks on the legends of the figures, which were not of sufficient quality. Legends of all figures and tables have been updated. We hope to have met the expectations of the reviewer 2. 

Figure 1 showing flow chart: each figure must have a full description about what it shows, so that it is clear to the reader as a stand-alone unit.
=> Legend of figure 1 has been updated : " Figure 1. Flow chart of the study. Thirty-five patients met the inclusion criteria: hospitalization in pediatric or neonatal intensive care unit, age: less than one year, aEEG monitoring reported in medical charts. Ten patients were subsequently excluded due to missing of the complete aEEG data set."

Table 1. The name of the table must clearly and in detail explain what the table shows. The characteristics of the patient should not be explained in the title of the table; that part should be separated and explained in the text.
Thank you for your remark. Legend of table 1 has been updated.

Figure 2.The name of the figure must clearly and in detail explain what it shows; with the ordinate and the abscissa, it should be written the number of patients and the chronological sequencein hours, not in the title of the figure.
We thank you for your comments. Figure 2 and its legend have been updated.

In supplement materials: abbreviations should be avoided in table names
Thank you for your valuable observation. Legends of supplemental figures and table have been updated.

Sincerely yours.